# Additive Effects of Solid Paraffins on Mechanical Properties of High-Density Polyethylene

**DOI:** 10.3390/polym15051320

**Published:** 2023-03-06

**Authors:** Asae Ito, Akid Ropandi, Koichi Kono, Yusuke Hiejima, Koh-hei Nitta

**Affiliations:** Polymer Physics Laboratory, Institute of Science and Engineering, Kanazawa University, Kanazawa 920-1192, Japan

**Keywords:** high-density polyethylene, solid paraffin, mechanical properties, dynamic mechanical spectra

## Abstract

In this work, two types of solid paraffins (i.e., linear and branched) were added to high-density polyethylene (HDPE) to investigate their effects on the dynamic viscoelasticity and tensile properties of HDPE. The linear and branched paraffins exhibited high and low crystallizability, respectively. The spherulitic structure and crystalline lattice of HDPE are almost independent of the addition of these solid paraffins. The linear paraffin in the HDPE blends exhibited a melting point at 70 °C in addition to the melting point of HDPE, whereas the branched paraffins showed no melting point in the HDPE blend. Furthermore, the dynamic mechanical spectra of the HDPE/paraffin blends exhibited a novel relaxation between −50 °C and 0 °C, which was absent in HDPE. Adding linear paraffin toughened the stress–strain behavior of HDPE by forming crystallized domains in the HDPE matrix. In contrast, branched paraffins with lower crystallizability compared to linear paraffin softened the stress–strain behavior of HDPE by incorporating them into its amorphous layer. The mechanical properties of polyethylene-based polymeric materials were found to be controlled by selectively adding solid paraffins with different structural architectures and crystallinities.

## 1. Introduction

High-density polyethylene (HDPE) is a typical semicrystalline polymer and has been widely used in various products, such as pipes, tanks, containers, sheets, and fibers, which were produced by various molding processes. The improvement of mechanical properties of HDPE-based materials makes it possible to expand their applicability. In a previous work, we investigated the additive effects of liquid paraffin (LP) on the supermolecular structure, such as spherulite and alternating lamellar structures, and dynamic mechanical spectra of HDPE [1]. In the plastic industry, adding LP to HDPE has been known to enhance the drawability of HDPE; thus, LP addition has been used as a technique for producing gel-spun polyethylene (PE) fibers and/or ultradrawing PE gels [2,3,4]. We found that LP is intensively concentrated in the amorphous phase of HDPE and expands its long period due to the thickening of the amorphous layer.

Dynamic mechanical spectra provide us with the fundamental characteristics of viscoelastic materials such as polymeric materials. It is well known that various PE-based materials have three relaxations in the temperature dependence of the dynamic mechanical moduli, identified conventionally as *α*, *β*, and *γ* in order of decreasing temperature. The *α* relaxation is observed in the range between room temperature and the melting temperature, *β* relaxation is observed around −20 to −30 °C, *γ* relaxation is observed at about −120 °C. The *α* relaxation is ascribed to the molecular motion in the crystalline phase and the *γ* relaxation is associated with the local molecular motion. The *β* relaxation is absent for HDPE with thin amorphous layers but is well identified for short-chain branched PE and ultra-high molecular weight PE with thick amorphous layers [5,6,7,8]. The *β* relaxation is sensitive to the molecular motion within amorphous layers. The existence of *β* relaxation strongly affects the mechanical properties such as toughening, impact resistance, and drawability.

A novel *β* processes in the range of −50 °C to −20 °C appear in the dynamic mechanical spectra of HDPE swollen with LP [1]. The *β* relaxation was due to the loosening of the tie chains that results from the thickening of the amorphous layer and mixing with floating LP chains, which inhibited the formation of a longer *trans* sequence of the tie chains between the lamellae [6]. Herein, we investigated the additive effects of solid paraffin, a crystalline compound and completely different from LP, to explore further modification methods for PE-based materials. Since LP exists as a liquid in HDPE while solid paraffin exists as microcrystals or crystalline, the behavior of these paraffins is expected to be quite different.

Solid paraffin is prepared by separating and refining the distillation fraction of crude oil, and it is a mixture of hydrocarbons with various isomeric structures [9,10,11]. When crude oil is separated and fractionated by weight through a vacuum distillation column, the main component separated and refined from relatively light oils with relatively lower boiling temperatures is normal paraffin wax (nPW). In addition, the main component refined from relatively heavy oils with relatively higher boiling temperatures is identified as a branched isotype hydrocarbon (iPW) and is called microwax (or microcrystalline wax). In this study, we compare the additive effects of these solid paraffin’s isomeric structures.

nPW is a powdery crystalline substance with a molecular weight of several hundreds and a carbon number distribution index of several tens. It exhibits high crystallizability and is called macrowax (or macrocrystalline wax). The average molecular weight of conventional macrowax is known to be around 300–500. Compared to nPW, iPW typically forms finer needle- or plate-like microcrystals and has a similar carbon number distribution. The crystallinity of iPW is lower than that of nPW owing to the presence of branches and many noncrystallizable components [12,13]. Notably, these two types of solid paraffin exhibit different physical properties: paraffin wax, i.e., macrowax, is translucent and brittle, and microwax is opaque and ductile [14]. Microwax usually has relatively higher molecular weight, smaller crystals such as microcrystals, and greater affinities to oil as compared to conventional paraffin waxes [9]. The average molecular weight of conventional microwax is known to be around 600 to 800 [10].

Solid paraffins have been used as a processing aid, particularly for PE plastics, and as a lubricant to prevent abrasion between polymer materials and molding machines [15]. Therefore, the additive effects of these solid paraffin compounds must be studied because, in practice, almost all products contain a small amount of solid paraffin. However, the effects of solid paraffin addition on the structural morphology and mechanical properties of the resulting PE solids after molding have not been sufficiently understood. This is because of a lack of systematic research on the additive effects of solid paraffins on the morphology and mechanical characteristics of HDPE in solids.

The additive effects of paraffin waxes with various melting temperatures and molecular weights on the mechanical and thermal properties of HDPE materials have been extensively investigated [16,17,18,19,20]. For example, Sotomayer et al. [16] investigated the stress–strain behavior of the blends of HDPE and solid paraffins with a carbon distribution of C18 to C50. They showed a lower drawability along with the solid paraffin content as compared to that of pure HDPE. Furthermore, the HDPE with 30 vol.% of solid paraffin showed the transition from ductile to brittle, and its yield point disappeared. However, there have been no studies focusing on the effects of the differences in structural architecture of paraffin waxes on the mechanical properties of PE. There have been several reports that the crystallizability depends on the species of solid paraffins: macrowax and microwax [21,22,23]. Herein, we separately investigated the effects of a normal paraffin wax (nPW) and a branched isotype paraffin wax (iPW) addition on the dynamic mechanical spectra and tensile properties of HDPE. The purpose of this work is to explore the possibility of using solid paraffins to modify the mechanical characteristics of HDPE.

## 2. Experiments

### 2.1. Sample Preparation

HDPE with a weight-average molecular weight of *M*_w_ = 1.86 × 10^5^ and a molecular weight distribution index of *M*_w_/*M*_n_ = 6.0 was used as the base material. Two types of solid paraffins with similar molecular weights were used as modifiers. One was a linear normal-type saturated hydrocarbon-rich solid paraffin (nPW) with 20–50 carbons. The other was an isotype saturated hydrocarbon-rich microwax (iPW) with 15–70 carbons. Both are commercially available (Nippon Seiro Co., Ltd., Tokyo, Japan).

The samples blended with HDPE were mixed with 5, 10, and 20 wt.% paraffin at 180 °C and 50 rpm in a twin screw-type mixer. Then, the pristine HDPE, HDPE/nPW, and HDPE/iPW samples were melt-pressed at 190 °C and 20 MPa for 5 min. The compression-molded sheets with a thickness of 200 ± 5 μm were prepared by quenching at 0 °C (in an ice water bath) for 5 min after being melted in the hot press. These sample sheets were used for the following measurements.

### 2.2. Measurements

Differential scanning calorimetry (DSC) measurements were conducted using a Diamond Differential Scanning Calorimeter (Diamond DSC, PerkinElmer, Waltham, MA, USA). The samples of about 3 mg weight sealed in aluminum pans were heated from −50 °C to 210 °C at a 20 °C/min heating rate under a nitrogen atmosphere. The crystallinity of the pristine HDPE sample was estimated from the DSC endothermic peak area. Table 1 summarizes the melting points of all samples.

The microstructure of the samples was investigated by small angle X-ray scattering (SAXS) and wide-angle X-ray diffraction (WAXD). SAXS measurements were performed at room temperature using a diffractometer (Nano-Viewer, Rigaku, Tokyo, Japan), with Cu K*α* X-ray (*λ* = 0.154 nm) at 40 mV/30 mA. The long period was estimated from the SAXS peak position after Lorentz correction according to Bragg’s law. Table 1 lists the long period values of all samples. WAXD measurements were conducted at room temperature using a Rigaku Nanoviewer—a nanoscale X-ray structure evaluation system for the solid paraffin blended with HDPE—with Cu K*α* X-ray (*λ* = 0.154 nm) at 40 mV/30 mA under room temperature. In addition, WAXD measurements of the pristine nPW and iPW waxes were conducted using a Rigaku Mini Flex II, with Cu K*α* X-ray (*λ* = 0.154 nm) at 30 kV/15 mA from 0° to 60° at a 2°/min scanning speed.

The spherulite morphology was investigated with small-angle light scattering (SALS) using the sample films prepared with about 50 μm thickness. SALS measurements were conducted at room temperature using a diode laser (*λ* = 532 nm) with 4.5 mW (CPS532, Thorlabs, NJ, USA) for initial HDPE, HDPE/nPW (80 w/20 w), and HDPE/iPW (80 w/20 w), and a He–Ne laser (*λ* = 633 nm) for HDPE/nPW (90 w/10 w), HDPE/nPW (95 w/5 w), HDPE/nPW (90 w/10 w), and HDPE/nPW (95 w/5 w). Light scattering pictures were taken using a CCD camera (DCC1545, Thorlabs, Newton, NJ, USA) under the perpendicular polarization (Hv) condition. The averaged spherulite radius was estimated from the maximum scattering angle using Stein’s equation [24]:(1)$$4\pi n\left(R/\lambda \right)\sin\left(\theta_{max}/2\right)=4.09$$

Here, *R* is the average radius of spherulite, *θ_max_* is the scattering angle, (*n* = 1.54) is the refractive index of PE. Table 1 presents the spherulite radii of all samples.

The dynamic mechanical measurements were conducted on 5-mm-wide and 30-mm-long strips of 200-μm-thick sample sheets. The gauge length between clamps was 20 mm, the frequency was 10 Hz, and the equipment used was DVE-V4 (UBM Co., Ltd., Kyoto, Japan). The temperature dependences of the storage modulus *E′*, loss modulus *E*″ and loss tangent tan *δ* were measured in the range from −150 °C to 150 °C at a heating rate of 2 °C/min under a nitrogen atmosphere.

Tensile tests were performed at room temperature using a tensile machine (Model 4466 INSTRON, Norwood, MA, USA) at a 10 mm/min elongation speed. Dumbbell-shaped specimens with a 10-mm gauge length cut from the 200-μm-thick sample sheets were used for the measurements. The tensile stress was determined from dividing the tensile load by the initial cross section and the tensile strain was calculated from the ratio of the increment of the length between clamps to the initial gauge length.

### 2.3. Characterization of the Starting Materials

Figure 1 shows the wide-angle X-ray diffraction (WAXD) profiles for HDPE, nPW, and iPW. The main crystal structure of HDPE and both solid paraffins was orthorhombic. The nPW exhibited considerably high crystallinity with almost no amorphous halo, whereas the iPW exhibited a broad amorphous halo, indicating low crystallizability [14,25].

Figure 2 illustrates the DSC curves of HDPE and solid paraffins. The melting points of HDPE, nPW, and iPW were 132 °C, 71 °C, and 43 °C, respectively. A slight shoulder was detected in the lower temperatures in nPW and iPW melting peaks, suggesting a solid–solid phase modification from orthorhombic to hexagonal crystals [26,27]. The crystallinity of the pristine HDPE sample estimated from the heat of fusion was 60%.

## 3. Results and Discussion

As well known, almost crystalline polymers prepared from the melts show spherulite structure which consists of crystalline lamellae composed of folded chain crystallites. The amorphous regions reside in the interlamellar regions and a number of tie chains linking between adjacent crystalline lamellae exists within the amorphous layers. Moreover, the spherulite is filled with the stacked lamellae in such a way that a constant degree of crystallinity is maintained. Figure 3 shows the pictures of the SALS patterns of the HDPE, HDPE/nPW (80 w/20 w), and HDPE/iPW (80 w/20 w) sheets. The similar scattering patterns were also observed for other blends. The four-leaf clover pattern reflects a typical spherulite structure. Based on the scattering patterns, the spherulite radius values were estimated to be about 2.0 μm, except for HDPE/nPW (80 w/20 w), which had a radius of 2.7 μm. The addition of solid paraffins limitedly affected the crystallization process of HDPE. However, a large amount of nPW enhanced the spherulite size, suggesting that nPW can act as secondary nuclei for spherulite growth. As another factor, Chen et al. [28] pointed out that the presence of 30% solid paraffin led to a PE crystal imperfection on the spherulite structure in HDPE due to the crystallinity and polarized optical microscopy.

Figure 4 summarizes the WAXD data for the HDPE blends. Orthorhombic crystals were observed in all samples. Moreover, adding both paraffins only slightly affected the diffraction patterns of the HDPE. Figure 5 shows the DSC curves for the HDPE as well as HDPE/nPW and HDPE/iPW blends. iPW addition did not affect the DSC curve of HDPE. However, in addition to the peak corresponding to the melting point of HDPE, nPW addition exhibited a slight peak at around 70 °C, corresponding to the melting point of nPW (Figure 5, inset). These results suggest that iPW showing lower crystallizability was not crystallized in the HDPE matrix. Conversely, nPW showing considerably higher crystallizability was partially crystallized in the HDPE matrix. The nPW nuclei or crystallites that evolved during blending may have caused HDPE spherulite growth through heterogeneous nucleation. The melting point of HDPE for both blends decreased slightly with increasing PW content (Table 1). The decrease in the melting point possibly resulted from a diluent effect according to Raoult’s law [29]. Consequently, it was suggested that the iPW molecules are completely dissolved in the amorphous phase of HDPE, whereas nPW molecules are also incorporated in the amorphous phases, but a part of nPW molecules forms crystalline domains.

The dynamic mechanical properties of semi-crystalline polymers are dominated by microscopic structural state such as crystalline form, crystallinity, crystalline lamellar thickness, amorphous layer thickness, and lamellar stacking [30,31,32]. Figure 6 depicts the dynamic mechanical spectra of HDPE, HDPE/nPW (80 w/20 w), and HDPE/iPW (80 w/20 w). Dynamic mechanical spectra of other blends are shown in the Appendix A as shown in Appendix A. The pristine HDPE showed two dispersions. The first dispersion at higher temperatures was attributed to the onset of the motion and migration of crystalline chains (*α* dispersion) within crystal lattices, appearing at about 50 °C toward the melting point. *α* relaxation is attributed to an interlamellar slip motion and to the chain motion along the c axis of crystal lattice. The second dispersion at lower temperatures shown in the pristine HDPE is *γ* dispersion, appearing at around −120 °C. The ascription of *γ* dispersion caused by local molecular motion was controversial [33,34,35,36,37,38,39] for the following reasons: microscopic motion involving disordered chains, crystal defects on the crystal surface, and a molecular glass transition resulting from the localized motion of amorphous chains [40].

Notably, *β* relaxation is not observed for typical HDPE materials. Linear low-density PE with short-chain branches, low density PE with long-chain branches, and ultrahigh molecular weight PE exhibited *β* relaxation in the range from −50 °C to 0 °C, because of an expansion of amorphous layer thickness [6]. These results led us to conclude that this dispersion was due to the inability of the tie chains linking the adjacent crystalline lamellae to take all the *trans* conformations, resulting in tie chain loosening [6]. Thus, the conformational molecular motion of the loose tie chains can be considered to be activated in temperature range from −50 °C to 0 °C. For HDPE, the amorphous layer thickness was very thin, leading the tie chains to an almost taut or all-*trans* conformation. Consequently, the taut tie chains linking between adjacent crystalline lamellae have no ability to activate the conformational motion prior to the onset of the crystalline chain motion (α dispersion). According to our previous study [1], we found that incorporating LP into the amorphous phase of HDPE produces a *β* relaxation peak and is capable of activating the molecular motion within the amorphous phases of HDPE.

A clear *β* relaxation peak was observed in the HDPE/iPW blends, similar to the additive effects of LP. The *β* relaxation peak reduced with decreasing solid paraffin contents (see Appendix A). Thus, this phenomenon results from mixing with solid paraffin compounds acting as floating and dangling chains within the amorphous phases of the HDPE matrix. For the HDPE/nPW blends, the *β* relaxation peak of *E*″ was broader than that of the iPW blends. Note that the concomitant decrease in *E*′ and *E*″ was around the melting point (70 °C) of nPW in the α dispersion region, which was also observed in the DSC (see Figure 5a). This implies that the melting of the crystallized nPW domains is reflected in the dynamic mechanical spectra. Consequently, the storage modulus *E*′ of HDPE/nPW was higher than those of HDPE and HDPE/iPW below the melting point of nPW (70 °C) since the crystallized nPW domains exist in the HDPE matrix (see Figure 6).

Figure 7 and Figure 8 illustrate the stress–strain curves at room temperature for all samples. Both curves exhibited stress–strain behavior for typical crystalline polymer solids. A clear yield peak appeared as a maximum in the stress–strain curve beyond the initial elastic region and is the onset point of temporary plastic deformation. Subsequently, the samples underwent a large-scale steady plastic flow due to necking, followed by hardening caused by increased stress [5,8]. The yield process is associated with the onset of irreversible deformation including the crystalline lamellar fragmentation and the morphological transformation of the spherulitic into the fibril structures. The yield stress corresponds to the failure strength and the area under a stress–strain plot up to the yield point is the failure toughness.

Interestingly, the samples exhibited softening with iPW addition because of the overall stress reduction keeping the elongation at break. This is likely because almost iPW molecules were incorporated into the amorphous phase of HDPE. Thus, the solid paraffin molecules within the amorphous phases activate the molecular motion of amorphous chains of HDPE, leading to softening the HDPE matrix. The softening effects are shown in Appendix A. Thus, the stress values of HDPE/iPW blends divided by the HDPE weight fraction are almost in accordance with the curve of pristine HDPE, indicating that adding iPW effectively softens the tensile properties.

In contrast, nPW addition enhanced yield behavior, inducing increased yield and necking stresses, as well as expanded the elongation at break. The yield behavior is caused by a release of the elastic energy stored in the crystalline lamellae due to the fragmentation of stacked crystalline lamellae under tensile deformation [41,42]. The enhancement of the yield stress will be due to that the stacked crystalline lamellae were reinforced by highly crystalline nPW components. These results indicate that the dual contributions of crystallized nPW domains and dissolved nPW in the amorphous phases enhance yield behavior as well as drawability.

## 4. Conclusions

In this work, we compared the additive effects of normal and iso-rich solid paraffin waxes on the structural morphology, dynamic mechanical characteristics, and uniaxial tensile properties of a HDPE. The addition of the two kinds of paraffin waxes limitedly affected the supermolecular structure, such as spherulites and alternating lamellar structure. When the amount of nPW was 20 wt.%, which is relatively higher amount, the spherulite size is enlarged. This suggests that nPW has a possibility to act as secondary nuclei for growing of spherulite of HDPE.

The addition of iso-rich, low-crystalline solid paraffin (iPW) exhibited a *β* relaxation in the range from −50 °C to 0 °C in the dynamic mechanical spectra, which is not observed in pristine HDPE. The appearance of a *β* relaxation peak suggests that solid paraffin compounds mixed with the amorphous chains of HDPE act as floating and dangling chains within the amorphous layers. Consequently, the overall stress level of the stress–strain curves degreased with the addition of iPW compounds. Moreover, the appearance of the relaxation in the −50 °C to 0 °C range can be expressed to improve the impact strength or the mechanical response under higher strain rates of HDPE materials. This leads to the overall softening of HDPE materials in the tensile properties. Thus, iPW compounds seem to act as a “softener” for HDPE. On the other hand, the addition of normal-type rich solid paraffins (nPWs) promoted not only the activation of amorphous chains in HDPE, but also the formation of crystallized components in the HDPE matrix, leading to the storage modulus *E’* of HDPE/nPW being higher than those of HDPE below the melting point of nPW (70 °C). As a consequence, the yield stress was increased, and the break point was expanded to higher strains by adding nPW. The dual contributions of the crystallized nPW components and the non-crystalline nPW components dissolved in the amorphous phases enhance yield behavior as well as drawability. Thus, nPW compounds seem to act as a “toughener” for HDPE.

Paraffin waxes, which are a mixture of these paraffins, have been widely used as lubricants to prevent the abrasion of the machine surface during molding. Considering that adding lubricants is indispensable for molding various polyolefin-based materials, lubricant selection on the basis of the structural type of the paraffin plays an important role in material production and application. In addition, investigating the additive effects of various paraffin compounds with identified structural architectures is imperative in designing PE-based materials with high mechanical performance. The modification effects of HDPE were found to depend largely on the structural architecture of the solid paraffin.

## Figures and Tables

**Figure 1 polymers-15-01320-f001:**
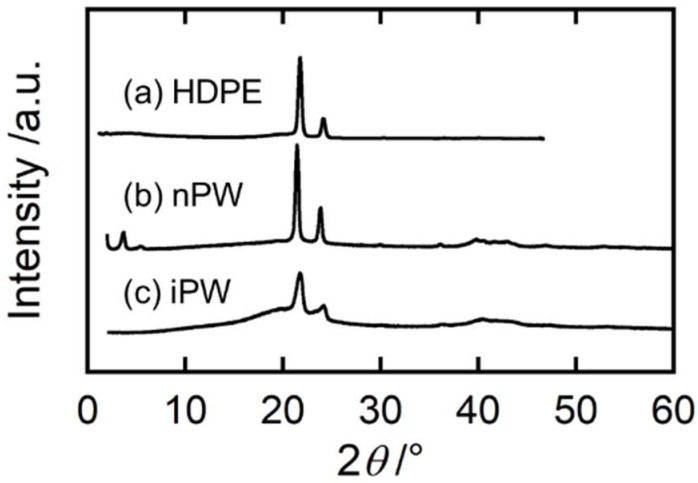
Wide-angle X-ray diffraction (WAXD) profiles of (**a**) HDPE, (**b**) nPW, and (**c**) iPW samples.

**Figure 2 polymers-15-01320-f002:**
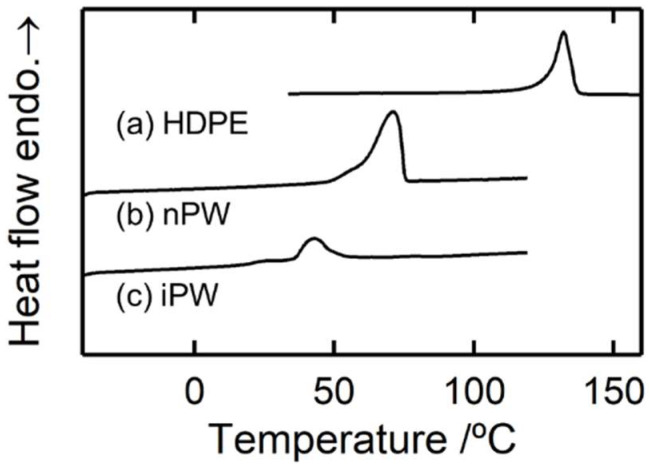
Differential scanning calorimetry (DSC) curves of (**a**) HDPE, (**b**) nPW, and (**c**) iPW samples.

**Figure 3 polymers-15-01320-f003:**
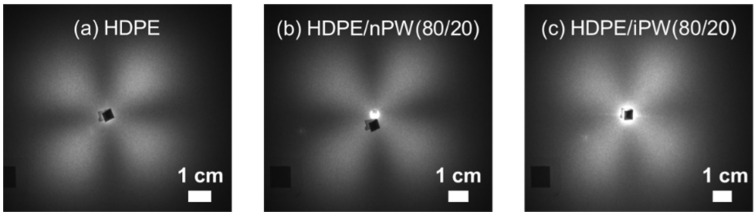
Small-angle light scattering (SALD) images of (**a**) HDPE, (**b**) the HDPE/nPW blend, and (**c**) the HDPE/iPW blend. The numerals in parentheses denote the wax weight fractions of HDPE and solid paraffins.

**Figure 4 polymers-15-01320-f004:**
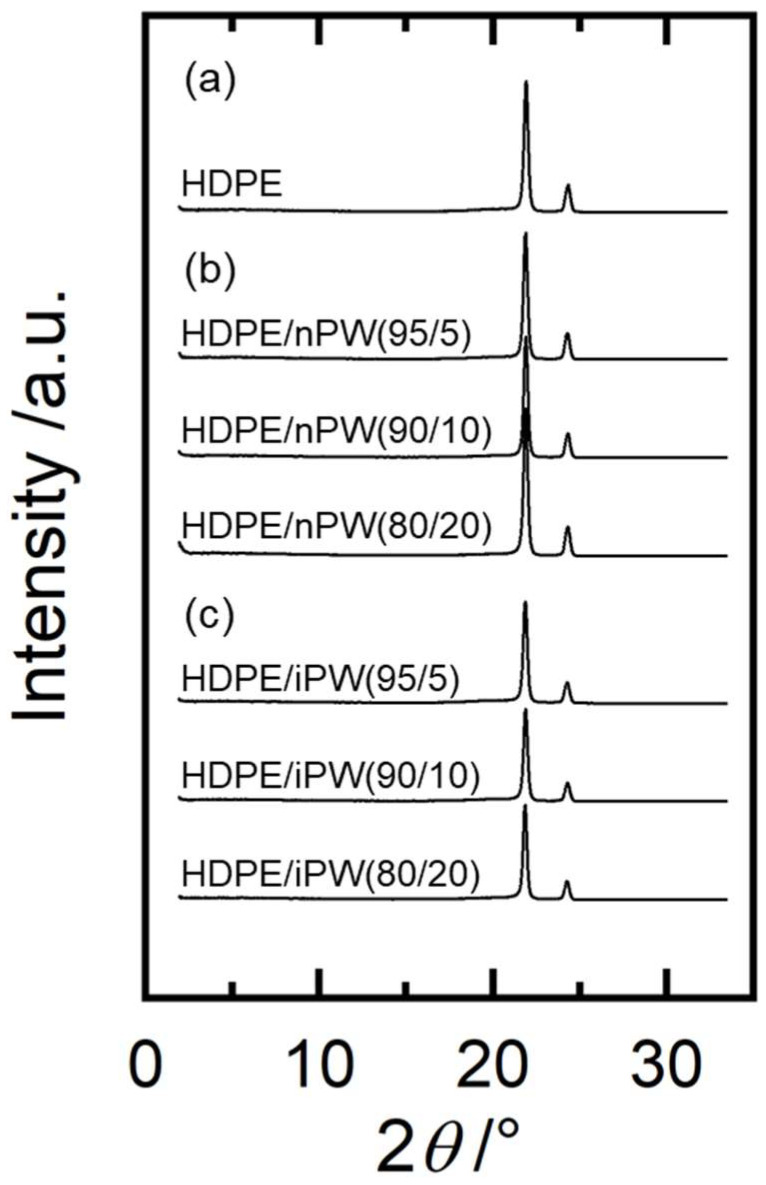
Wide-angle X-ray diffraction (WAXD) profiles of (**a**) HDPE, (**b**) the HDPE/nPW blends, and (**c**) the HDPE/iPW blends. The numerals in parentheses denote the wax weight fractions of HDPE and solid paraffins.

**Figure 5 polymers-15-01320-f005:**
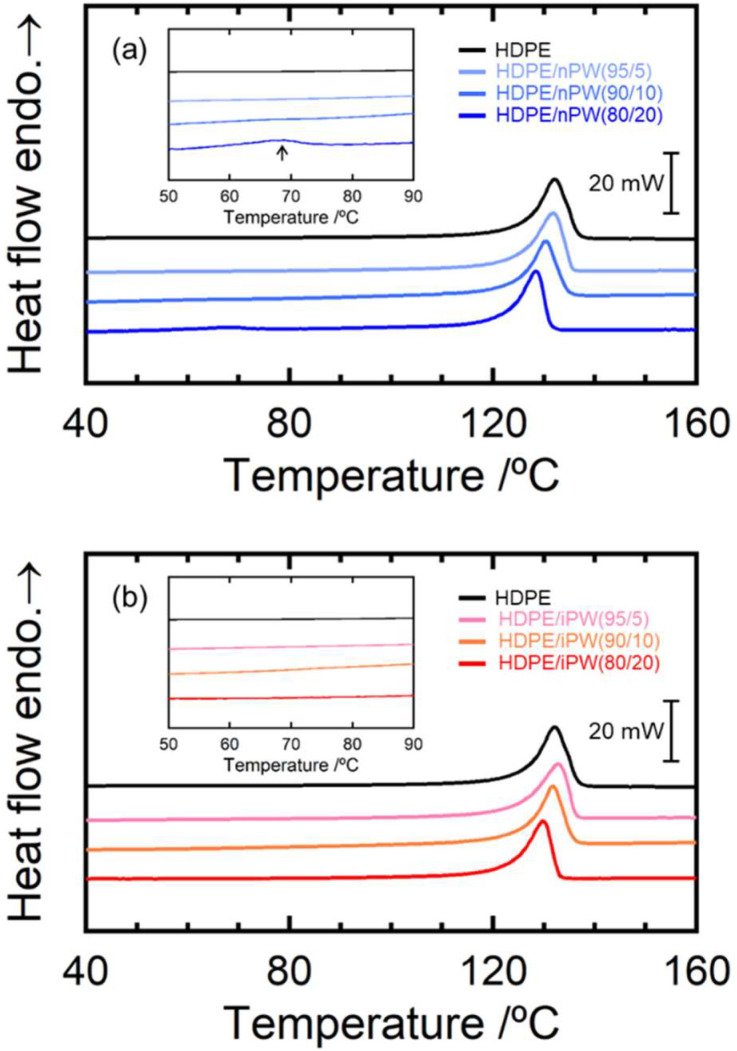
Differential scanning calorimetry (DSC) curves of the (**a**) the HDPE/nPW blends and (**b**) the HDPE/iPW blends. The numerals in parentheses denote the weight fractions of HDPE and solid paraffins.

**Figure 6 polymers-15-01320-f006:**
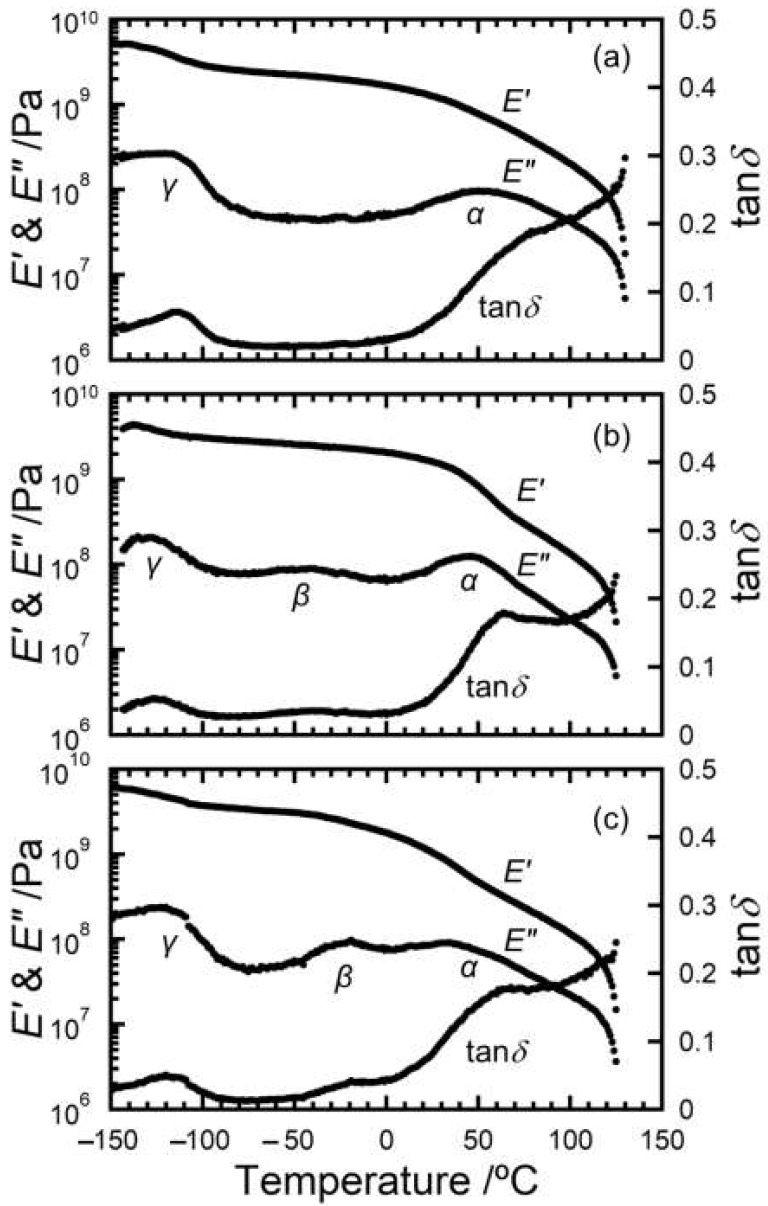
Dynamic mechanical spectra of (**a**) HDPE, (**b**) the HDPE/nPW (80 w/20 w) blend, and (**c**) the HDPE/iPW (80 w/20 w) blend. *E*′ is the storage modulus, *E*″ is the loss modulus, and tan *δ* is the loss tangent.

**Figure 7 polymers-15-01320-f007:**
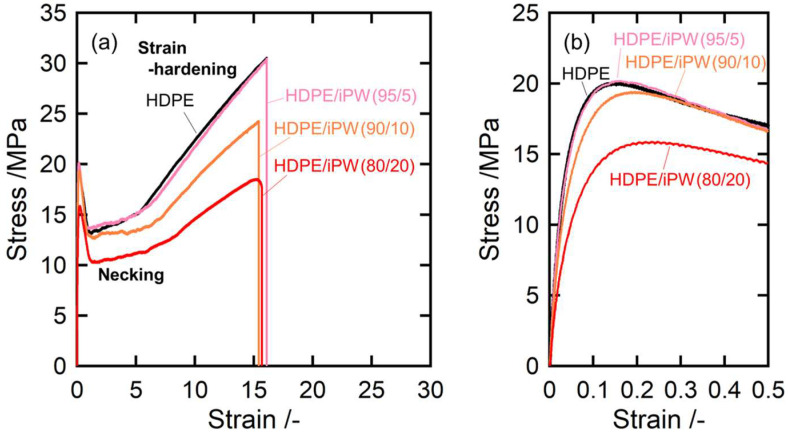
Stress–strain curves of HDPE and the HDPE/iPW blends: (**a**) overall view of the stress–strain curves and (**b**) the stress–strain curves in the yield deformation region. The numerals in parentheses denote the weight fractions of HDPE and solid paraffins.

**Figure 8 polymers-15-01320-f008:**
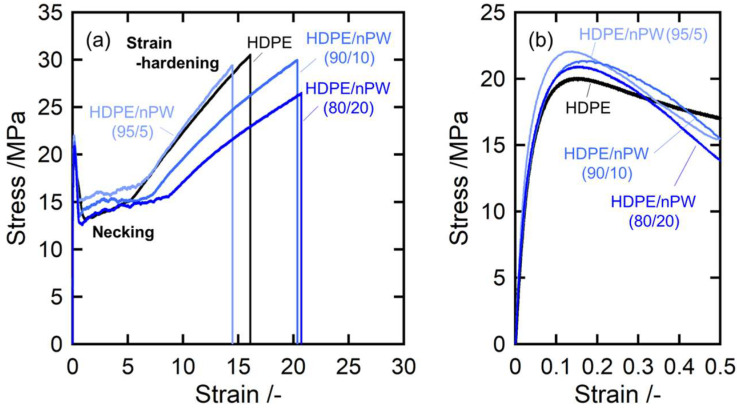
Stress–strain curves of HDPE and the HDPE/nPW blends: (**a**) overall view of the stress–strain curves and (**b**) the stress–strain curves in the yield deformation region. The numerals in parentheses denote the weight fractions of HDPE and solid paraffins.

**Table 1 polymers-15-01320-t001:** Characteristics of HDPE, HDPE/nPW, and HDPE/iPW.

Sample Code	*L*_p_/nm	*R*/μm	*T*_m_/°C
HDPE	26	2.0	132
HDPE/nPW (95 w/5 w)	26	1.9	132
HDPE/nPW (90 w/10 w)	26	1.8	130
HDPE/nPW (80 w/20 w)	27	2.7	128
HDPE/iPW (95 w/5 w)	26	1.8	133
HDPE/iPW (90 w/10 w)	26	1.9	132
HDPE/iPW (80 w/20 w)	27	1.9	130

*L*_p_: Long period; *R*: Spherulite radius; *T*_m_: Melting point.

## Data Availability

The data that support the findings of the study are available on the request from the corresponding author.

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
