# Peer review of "Additive Effects of Solid Paraffins on Mechanical Properties of High-Density Polyethylene"

_polymers, 2023, doi:10.3390/polym15051320_

Round 1
Reviewer 1 Report
Introduction:
Why do authors add paraffin into HDPE? or Usually what’s the function of pafaffin wax in plastic production? It need to explain.
Both liquid and solid paraffin have been widely used in industry. What’s new of this article? It needs to explain. Replacement of liquid with solid paraffin is not enough to support.
How about other researcher’s study on this topic ? The authors should give more literatures and analysis.
Methods
More information need to know about IPW and wPW. These information are critical to explain the results.
Size of the sheet should be given.
The authors used too much paraffin wax in HDPE. Usually it should not exceed 1 part when HDPE take 100 parts.
English need to improve.
Author Response
We have attached the letter to reviewers.

Reviewer 2 Report
The presented manuscript builds on the authors' original work and evaluates the mechanical properties of high-density polyethylene (HDPE), mainly focusing on the stress–strain behavior. The authors investigate the additive effects of solid paraffin, adding of the two types of nPW and the iPW with 5, 10, and 20 wt.% on the dynamic mechanical spectra and tensile properties of HDPE.
For the completeness of the text, I have the following comments:
* The abstract is too short and general. Complete the basic HDPE characterization methods and partial results.
* The text lacks a reference to Table 2. The values of this table are not commented on in the manuscript text.
* I consider the description of the results of only selected HDP, HDPE/nPW (80w/20w) and 116 HDPE/iPW (80w/20w) samples to be a major shortcoming of the publication. A more detailed description of the characteristics of the other samples is completely missing, while there are differences in the individual measurement results. This needs to be worked out.
* In the conclusion, it is stated that "the additive effects of normal- and iso-rich solid paraffins on the morphology were investigated". The HDPE morphology is not described in the manuscript ...
Author Response
We have attached the letter to reviewer2.

Reviewer 3 Report
The authors studied the effects of adding two different types of paraffins (linear and branched) into high-density polyethylene regarding their viscosity and mechanical behaviors. Characterization methods such as DSC, SAXS, WAXD, SALS were utilized. In general, this is an interesting paper. However, the following issues/questions needed to be addressed before further actions:
- Between line 19-21, the authors claimed the novel relaxation of HDPE (beta relaxation). However, I believe this has been discussed previously. For instance: https://doi.org/10.1016/j.polymertesting.2010.03.005
- For the additives used within the study, the nPW has Mw 300-500 while iPW has Mw 450-700, which are pretty different. Do the authors think this difference can lead to the different behaviors of the systems? If not, please provide some more evidences support the argument.
- For the DSC results in Figure 2, the authors claimed the shoulder was due to the phase transition. However, could it be the melting peak of the low Mw polymers given the high PDI of the additives used within this study?
- Since linear and branched PW were used within this study, the effects of entropy differences caused by the branched structure should be included and discussed within this study. For instance: https://doi.org/10.1021/acs.macromol.9b01801; https://doi.org/10.1002/polb.1995.090331709; https://doi.org/10.1021/acs.macromol.8b02242 have discussed the branched structure can lead to the uneven distribution of the additives within the blends.
- The authors claimed adding nPW will enhanced the mechanical performance of the blends. However, within Figure 7a, by adding 5% nPW, the blend showed worse mechanical performance than the HDPE itself. Please comment on it.
- For Figure 6, only the ratio of 80/20 were shown within the manuscript. I wonder if other ratios also exhibit similar results? Please add the data into the supporting information to support the claims.
- In line 164, the authors claimed no beta relaxation can be observed for typical HDPE materials. However, I believe if the HDPE was stretched to a specific condition, the beta relaxation can also be observed. Please correct me if I was wrong and comment on it.
- I would like to suggest the authors include a paragraph about how the newly observed beta relaxation will affect the application of HDPE.
Author Response
We have attached the letter to reviewer 3.

Round 2
Reviewer 2 Report
The authors made revisions in the areas required. The manuscript is acceptable in a revised form.
Reviewer 3 Report
The authors have addressed all my questions and I do not have any further comments.